**Data Availability Statement:** The data described in the manuscript will be made available upon request, application and approval (Kanazawa

# Protein intake in inhabitants with regular exercise is associated with sleep quality: Results of the Shika study

**Fumihiko Suzuki**[1]*, **Emi Morita**[2,3], **Sakae Miyagi**[4], **Hiromasa Tsujiguchi**[1,5], **Akinori Hara**[1,5], **Thao Thi Thu Nguyen**[1,6], **Yukari Shimizu**[1,7], **Koichiro Hayashi**[1,8], **Keita Suzuki**[1,8], **Takayuki Kannon**[5,9], **Atsushi Tajima**[5,9], **Sumire Matsumoto**[2,10], **Asuka Ishihara**[2,10], **Daisuke Hori**[11], **Shotaro Doki**[11], **Yuichi Oi**[11], **Shinichiro Sasahara**[11], **Makoto Satoh**[2], **Ichiyo Matsuzaki**[2,11], **Masashi Yanagisawa**[2], **Toshiharu Ikaga**[12], **Hiroyuki Nakamura**[1,5]

1 Department of Environmental and Preventive Medicine, Graduate School of Medical Science, Kanazawa University, Kanazawa, Ishikawa, Japan, 2 International Institute for Integrative Sleep Medicine (WPI-IIIS), University of Tsukuba, Tsukuba, Ibaraki, Japan, 3 Forestry and Forest Products Research Institute, Forest Research and Management Organization, Tsukuba, Ibaraki, Japan, 4 Innovative Clinical Research Center, Kanazawa University, Kanazawa, Ishikawa, Japan, 5 Advanced Preventive Medical Sciences Research Center, Kanazawa University, Kanazawa, Ishikawa, Japan, 6 Faculty of Public Health, Haiphong University of Medicine and Pharmacy, Ngo Quyen, Hai Phong, Vietnam, 7 Department of Nursing, Faculty of Health Sciences, Komatsu University, Komatsu, Ishikawa, Japan, 8 Department of Public Health, Graduate School of Advanced Preventive Medical Sciences, Kanazawa University, Kanazawa, Ishikawa, Japan, 9 Department of Bioinformatics and Genomics, Graduate School of Advanced Preventive Medical Sciences, Kanazawa University, Kanazawa, Ishikawa, Japan, 10 Ph.D. Program in Human Biology, University of Tsukuba, Tsukuba, Ibaraki, Japan, 11 Faculty of Medicine, University of Tsukuba, Tsukuba, Ibaraki, Japan, 12 School of Science for Open and Environmental Systems, Graduate School of Science and Technology, Keio University, Kohoku, Yokohama, Japan

* f-suzuki@stu.kanazawa-u.ac.jp

## Abstract

### Study objectives

Although associations between sleep quality and environmental factors and nutrient intake have been reported, interactions between these factors have not been elucidated in detail. Therefore, this cross-sectional study examined the effects of regular exercise and nutrient intake on sleep quality using the Pittsburgh Sleep Quality Index (PSQI), which is the most frequently used index for sleep evaluation.

### Methods

The participants included 378 individuals aged 40 years or older living in Shika Town, Ishikawa Prefecture. Of these individuals, 185 met the inclusion criteria. The participants completed a self-administered questionnaire assessing lifestyle habits and frequency and duration of exercise, the PSQI, and the brief-type self-administered diet history questionnaire (BDHQ) on nutrient intake.

### Results

A two-way analysis of covariance on regular exercise and PSQI scores indicated that protein intake (17.13% of energy) was significantly higher in the regular exercise and PSQI ≤10

University Ethics Committee. Person in charge: Yuko Katsuragi <pub-jim2@staff.kanazawa-u.ac. jp>).

**Funding:** The present study was supported by a Grant-in-Aid for Scientific Research (B) by the Ministry of Education, Culture, Sport, Science and Technology (MEXT), number 15H04783 and 16H03245. The funders had no role in study design, data collection and analysis, decision to publish, or preparation of the manuscript.

**Competing interests:** The authors have declared that no competing interests exist.

groups than in the non-regular exercise or PSQI $\geq 11$ groups ($p = 0.002$). In a multiple logistic regression analysis with PSQI scores ($\leq 10$ and $\geq 11$), protein intake was a significant independent variable in any of the models adjusted for confounding factors such as age, sex, body mass index, current smoker, and current drinker (OR: 1.357, 95% CI: 1.081, 1.704, $p = 0.009$) in the regular exercise group but not in the non-regular exercise group.

### Conclusions

We identified a positive relationship between sleep quality and protein intake in the regular exercise group. These findings suggest that regular exercise at least twice a week for 30 minutes or longer combined with high protein intake contributes to good sleep quality.

## Introduction

Sleep plays an important role in maintaining health. Sleep disorders have been shown to negatively affect lifestyle-related diseases, such as metabolic syndrome [1], hypertension [2, 3], diabetes [4, 5], and cardiovascular disease [6, 7]. The following underlying mechanisms have been suggested for this association: neurobiological and physiological stressors [2]; the inhibition of glycemic control, which increases dietary intake through the secretion of ghrelin [5]; and the promotion of insulin resistance by increasing cortisol, IL-6, and TNFα levels [5]. Therefore, preventing sleep disorders is important for maintaining health.

Exercise has been noted to have a positive impact on sleep disorders. Epidemiological studies on sleep disorders and exercise/physical activity have examined non-restorative sleep in middle-aged and elderly individuals [8] and sleep quality in nursing homes [9]. Intervention studies have reported improvements in sleep quality in adults [10] and elderly individuals with depression [11]. Exercise affects sleep via the following mechanisms: its thermoregulatory effects reduce wake times during the night [10], it facilitates sleep onset by activating a heat dissipation mechanism controlled by the hypothalamus to increase central body temperature [12], and it improves mood due to its antidepressant/anxiety effects [13]. In animal studies, exercise was shown to increase the levels of adenosine, which activates the sleep center in the hypothalamus [14], and serotonin, which synthesizes the sleep hormone melatonin [12].

Nutrition has been investigated as another factor related to sleep disorders. Epidemiological studies have reported a relationship between sleep quality and micronutrients such as carotenoids [15], vitamin $B_{12}$ [15, 16], calcium [17], and selenium [18]. However, the relationships between sleep and macronutrients remain unclear. A previous study indicated the presence of a relationship between sleep and protein intake [16], but another study reported that no such relationship existed [19]. The latter study demonstrated that sleep was associated with lipid and carbohydrate intake [19]. Thus, the findings obtained from previous epidemiological studies are inconsistent. These discrepancies may exist because of the lack of a uniform method for evaluations using questionnaires [15, 16, 18, 20, 21]. Therefore, we conducted an epidemiological study to investigate the factors affecting sleep using the Pittsburgh Sleep Quality Index (PSQI), which is one of the most frequently used indices for evaluating self-rated sleep quality in sleep medicine.

The role of environmental factors must be considering when examining the association between nutrient intake and sleep. However, previous studies have not yet investigated this triangular relationship in detail. Two previous studies that reported different findings on the relationship between protein intake and sleep [16, 19] did not perform an analysis adjusted for environmental factors such as exercise. Therefore, the effects of interactions between environmental factors and nutrient intake on sleep remain unclear. The present cross-sectional study examined the effects of regular exercise and nutrient intake on sleep quality.

## Methods

### Data collection

In this cross-sectional study, comprehensive health survey data were collected from the residents of Shika Town, Ishikawa Prefecture, Japan, between November 2017 and February 2018. As of November 2017, there were 21,007 residents in Shika Town, and 15,012 were older than 40 years [22]. The Shika study epidemiologically investigates the causes of lifestyle-related diseases through interviews, self-administered questionnaires, and comprehensive medical examinations. Previous studies have also examined the relationship between nutrition and health [23–25].

### Participants

This study was conducted on participants recruited from those who underwent a medical examination in Shika Town. For details, a total of 378 people aged 40 years and older who live in four model districts (Horimatsu, Tsuchida, Higashimatsudo, and Togi) provided their consent to participate in this sleep study. Of these individuals, 193 were excluded because they did not meet the survey criteria [169 participants did not complete the brief-type self-administered diet history questionnaire (BDHQ), 1 participant did not have energy records within 600 – 4000Kcal/day, and 23 participants did not complete the smoking, drinking, or exercise questionnaire]. Fig 1 shows the inclusion criteria. In total, 185 participants (95 males and 90 females; mean age ± standard deviation: 60.5 ± 9.7 years, range 41–83 years) who answered all relevant questions in the questionnaires and did not withdraw their consent were included in the analysis.

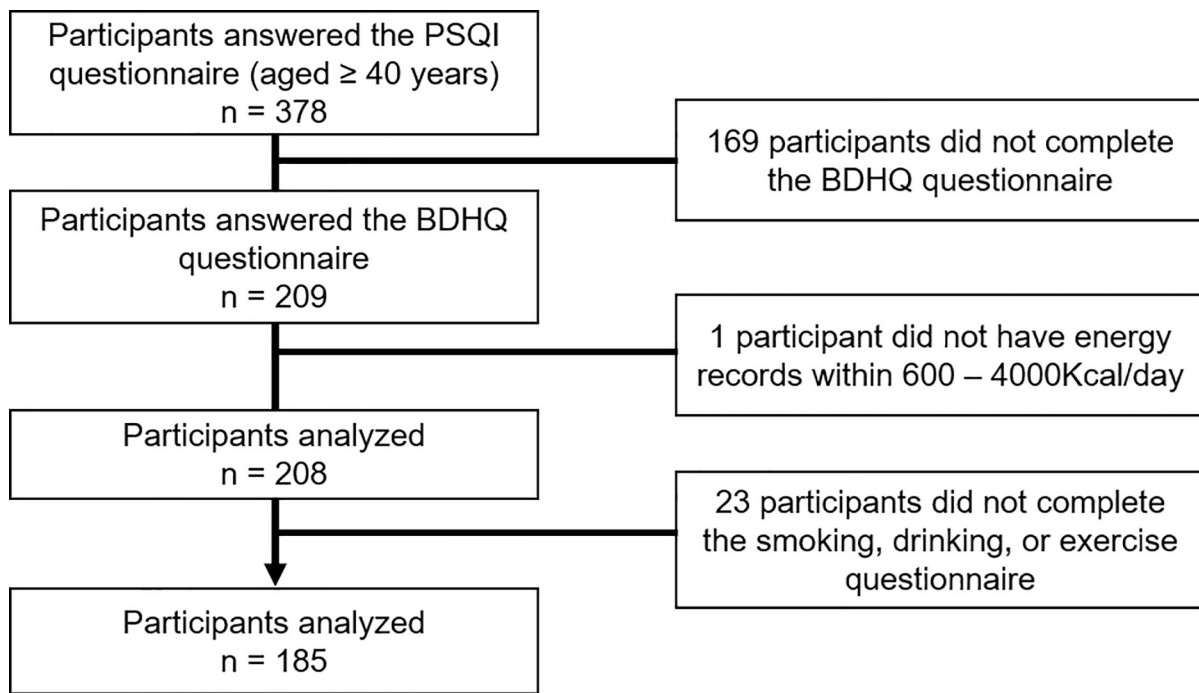

**Fig 1. Participant recruitment chart.** * This reference value was chosen for the following reasons: less than 600 kcal/day is equivalent to half the energy intake required for the lowest physical activity category; more than 4000 kcal/day is equivalent to 1.5 times the energy intake required for the medium physical activity category.

## Questionnaire and measurements

The participants completed a self-administered questionnaire on lifestyle and underlying diseases. Lifestyle items included the number of exercise days per week and the mean exercise time during each session, whether they were current smokers (1. yes, 2. no) and/or current drinkers (1. yes, 2. no), and education (1. junior high school, 2. high school 3. junior college, 4. university or higher). Underlying disease items included metabolic syndrome (1. yes, 2. no), hypertension (1. yes, 2. no), diabetes (1. yes, 2. no), angina (1. yes, 2. no), myocardial infarction (1. yes, 2. no), and depression (1. yes, 2. no). Body mass index (BMI) was measured using health survey data from the Shika study.

Nutrient intake was assessed using the BDHQ [26, 27]. The BDHQ is a four-page structured questionnaire that assesses the consumption frequency of 58 foods and beverages that are commonly consumed by the general Japanese population. The BDHQ estimates dietary intake in the last month using an *ad hoc* computer algorithm. The validity of the BDHQ has been demonstrated in previous studies in Japanese populations [26, 27]. To analyze nutrient data, the density method was used to estimate intake per 1000 Kcal. The following formula was used to calculate the energy intake ratio (% energy) of energy-producing nutrients (proteins, lipids, carbohydrates, and alcohol): energy intake from each nutrient/energy intake (EN) × 100, adjusted intake of non-energy-producing nutrients: crude intake of various nutrients/ EN × 1000 Kcal.

Sleep status was assessed using the PSQI [28, 29]. The PSQI assesses sleep quality and disturbances over a one-month period and consists of the following components: subjective sleep quality, sleep latency, sleep duration, habitual sleep efficiency, sleep disturbance, the use of sleeping medication, and daytime dysfunction. Each component of the PSQI is scored from 0 to 3. The PSQI global score is a sum of these components that ranges between 0 and 21, with higher scores indicating poorer sleep quality. The validity of the PSQI has been demonstrated in previous studies in Japanese populations [29].

## Statistical analysis

Regular exercise was defined as exercise for at least 30 minutes at a time and twice a week [30]. Buysse *et al.* [28] defined a PSQI score >5 as poor sleep quality; however, their study population included adolescents and differed from that in our Shika study, which included those aged 40 years and older. In the present study, the median PSQI of the participants was 10; thus, they were classified into PSQI ≤10 and ≥11 groups. The age distribution of the participants in the present study did not differ from that of the Shika Town inhabitants.

The distribution of variables was checked by the Kolmogorov–Smirnov, and Shapiro–Wilk normality tests, or the normal distribution curve in the histogram was confirmed before using other statistical tests. The Student's *t*-test was used to compare the means of continuous variables and the Chi-square test was performed to compare the proportions of categorical variables. All participants were stratified into two groups based on their PSQI scores (PSQI ≤10 and ≥11) and whether they participated in regular exercise (regular exercise group and non-regular exercise group). A two-way analysis of covariance (ANCOVA) was used to examine the effects of the interaction between regular exercise and PSQI on nutrient intake. The following confounding factors were adjusted for: age, sex, BMI, current smoker, current drinker, education, hypertension, and diabetes. A multiple logistic regression analysis was conducted to examine the effects of regular exercise and nutrient intake on sleep quality. The dependent variable was the PSQI (≤10 and ≥11). We used three models in the logistic regression analysis. Model 1 included the individual factors of age, gender, and BMI. Model 2 included the

environmental factors of current smoker and current drinker together with individual factors. Model 3 included the disease factors of hypertension and diabetes together with individual factors. Additionally, the analyses were stratified by whether the participants performed regular exercise. Pearson's correlation coefficient was used to confirm multicollinearity. Specifically, there was no value of $|r| > 0.9$ in the correlation matrix table between independent variables. The forced input method was used for variable selection. The significance level was set at 5%. IBM SPSS Statistics version 25 for Windows (IBM, Armonk, NY, USA) was used for the statistical analysis.

### Ethics statement

The present study was conducted with the approval of the Ethics Committee of Kanazawa University (No. 1491). Written informed consent was obtained from all participants.

## Results

### Participant characteristics

The participants' sleep quality and nutrient intakes are shown in Table 1. Among the 185 participants, there were 95 males with a mean age of 60.9 years ($SD = 9.6$) and 90 females with a mean age of 60.3 years ($SD = 10.1$); there was no significant difference between genders. BMI ($p < 0.001$) was significantly higher in males than in females. Significantly more males were current smokers ($p < 0.000$), current drinkers ($p < 0.001$), and had metabolic syndrome ($p < 0.001$), diabetes ($p = 0.049$), and angina ($p = 0.019$) than females. The mean PSQI was 10.7 ($SD = 2.7$) in males and 10.7 ($SD = 2.8$) in females, with no significant difference between genders. When comparing nutrients, the total energy ($p < 0.001$) was significantly higher in males. Conversely, the intakes of other nutrients, excluding carbohydrates, sodium, vitamin D, and vitamin $B_{12}$, were significantly higher in females.

### Comparison with the PSQI

The mean age of the 106 participants in the PSQI $\leq 10$ group was 59.2 years, which was significantly younger than that of the 79 participants in the PSQI $\geq 11$ group (62.4 years, $p = 0.026$) (Table 2). The PSQI $\leq 10$ group reported significantly more exercise days per week ($p = 0.004$) and a significantly longer mean exercise time per session ($p = 0.008$). Furthermore, these factors were significantly greater in the PSQI $\leq 10$ group even after adjusting for age, sex, BMI, current smoker, current drinker, education, hypertension, and diabetes (exercise days and exercise time: $p < 0.001$ and $p = 0.004$, respectively). Therefore, regular exercise was beneficial for sleep quality. The proportion of participants with metabolic syndrome ($p = 0.026$) was significantly higher in the PSQI $\leq 10$ group. When comparing each nutrient, the intakes of retinol equivalent ($p = 0.044$) and vitamin $B_2$ ($p = 0.024$) were significantly higher in the PSQI $\leq 10$ group than in the PSQI $\geq 11$ group.

### Comparisons with regular exercise

The mean age of the 59 participants in the regular exercise group (64.2 years) was significantly older than that of the 126 participants in the non-regular exercise group (58.8 years, $p < 0.001$) (Table 3). The mean PSQI ($p = 0.003$) was significantly higher in the non-regular exercise group. When comparing each nutrient, the intakes of protein ($p < 0.001$), minerals ($p < 0.001$), and 12 kinds of vitamins were significantly higher in the regular exercise group than that in the non-regular exercise group.

**Table 1. Participant characteristics.**

| | Total (N = 185) | | Male (N = 95) | | Female (N = 90) | | p Value * |
|---|---|---|---|---|---|---|---|
| | Mean (*n*) | SD (%) | Mean (*n*) | SD (%) | Mean (*n*) | SD (%) | |
| Age, years | 60.5 | 9.7 | 60.9 | 9.6 | 60.3 | 10.1 | 0.791 |
| BMI, kg/m$^2$ | 23.1 | 3.1 | 23.8 | 3.3 | 22.2 | 2.6 | <**0.001** |
| Exercise / week, days | 1.5 | 2.3 | 1.4 | 2.4 | 1.5 | 2.2 | 0.700 |
| Exercise time / each session, minutes | 27.2 | 52.2 | 24.4 | 53.4 | 30.2 | 51.0 | 0.452 |
| Current smoker, n (%) | 31 | 16.8 | 25 | 26.3 | 6 | 6.7 | <**0.001** |
| Current drinker, n (%) | 107 | 57.8 | 75 | 78.9 | 32 | 35.6 | <**0.001** |
| Education | | | | | | | 0.142 |
| Junior high school, n (%) | 38 | 20.5 | 19 | 20.0 | 19 | 21.1 | |
| High school, n (%) | 79 | 42.7 | 39 | 41.1 | 40 | 44.4 | |
| Junior college, n (%) | 39 | 21.1 | 15 | 15.8 | 24 | 26.7 | |
| University or higher, n (%) | 29 | 15.7 | 22 | 23.2 | 7 | 7.8 | |
| PSQI | 10.7 | 2.7 | 10.7 | 2.7 | 10.7 | 2.8 | 0.988 |
| **Underlying diseases** | | | | | | | |
| metabolic syndrome, n (%) | 46 | 24.9 | 36 | 37.9 | 10 | 11.1 | <**0.001** |
| Hypertension, n (%) | 61 | 32.8 | 37 | 38.5 | 24 | 26.5 | 0.086 |
| Diabetes, n (%) | 14 | 8.0 | 11 | 12.1 | 3 | 3.6 | **0.049** |
| Angina, n (%) | 10 | 5.4 | 9 | 9.9 | 1 | 1.2 | **0.019** |
| Myocardial infarction, n (%) | 3 | 1.6 | 2 | 2.2 | 1 | 1.2 | 1.000 |
| Depression, n (%) | 1 | 0.6 | 1 | 1.1 | 0 | 0.0 | 1.000 |
| **Nutrients** | | | | | | | |
| Total energy, Kcal | 1968.21 | 632.39 | 2214.45 | 648.63 | 1708.29 | 499.40 | <**0.001** |
| Protein, %energy | 15.11 | 3.08 | 14.33 | 3.12 | 15.93 | 2.84 | <**0.001** |
| Fat, % energy | 25.36 | 5.76 | 23.65 | 5.91 | 27.17 | 5.04 | <**0.001** |
| Carbohydrate, % energy | 53.66 | 7.58 | 53.08 | 7.97 | 54.27 | 7.15 | 0.287 |
| minerals, % energy | 10.32 | 1.98 | 9.72 | 1.89 | 10.94 | 1.88 | <**0.001** |
| Sodium, mg/1000 Kcal | 2397.76 | 486.77 | 2338.00 | 493.78 | 2460.84 | 473.82 | 0.086 |
| Potassium, mg/1000 Kcal | 1413.49 | 412.08 | 1252.45 | 331.72 | 1583.49 | 421.83 | <**0.001** |
| Calcium, mg/1000 Kcal | 291.66 | 109.87 | 260.60 | 109.61 | 324.45 | 100.74 | <**0.001** |
| Magnesium, mg/1000 Kcal | 141.30 | 32.01 | 130.51 | 28.43 | 152.68 | 31.76 | <**0.001** |
| Phosphorus, mg/1000 Kcal | 573.06 | 129.32 | 537.85 | 132.97 | 610.22 | 114.81 | <**0.001** |
| Iron, mg/1000 Kcal | 4.23 | 1.05 | 3.86 | 0.88 | 4.61 | 1.08 | <**0.001** |
| Zinc, mg/1000 Kcal | 4.47 | 0.64 | 4.28 | 0.68 | 4.67 | 0.53 | <**0.001** |
| β-carotene equivalent, μg/1000 Kcal | 2044.46 | 1278.57 | 1630.35 | 975.87 | 2481.58 | 1413.37 | <**0.001** |
| Retinol equivalent, μg/1000 Kcal | 359.43 | 175.78 | 325.33 | 177.92 | 395.43 | 167.03 | **0.006** |
| Vitamin D, μg/1000 Kcal | 8.17 | 5.23 | 7.83 | 5.33 | 8.53 | 5.12 | 0.363 |
| α-Tocopherol, mg/1000 Kcal | 3.96 | 1.06 | 3.58 | 0.91 | 4.37 | 1.07 | <**0.001** |
| Vitamin K, μg/1000 Kcal | 168.40 | 87.03 | 142.38 | 67.30 | 195.85 | 96.90 | <**0.001** |
| Vitamin B1, mg/1000 Kcal | 0.41 | 0.09 | 0.38 | 0.08 | 0.45 | 0.09 | <**0.001** |
| Vitamin B2, mg/1000 Kcal | 0.66 | 0.17 | 0.61 | 0.16 | 0.72 | 0.16 | <**0.001** |
| Niacin, mg/1000 Kcal | 9.47 | 2.49 | 8.90 | 2.36 | 10.08 | 2.49 | **0.001** |
| Vitamin B6, mg/1000 Kcal | 0.70 | 0.17 | 0.65 | 0.15 | 0.75 | 0.18 | <**0.001** |
| Vitamin B12, μg/1000 Kcal | 5.53 | 2.75 | 5.31 | 2.77 | 5.77 | 2.72 | 0.258 |
| Folic acid, μg/1000 Kcal | 178.61 | 63.21 | 158.26 | 50.75 | 200.09 | 68.06 | <**0.001** |
| Pantothenic acid, mg/1000 Kcal | 3.40 | 0.72 | 3.13 | 0.66 | 3.69 | 0.67 | <**0.001** |

(*Continued*)

**Table 1.** (Continued)

| | Total (N = 185) | | Male (N = 95) | | Female (N = 90) | | p Value * |
|---|---|---|---|---|---|---|---|
| | Mean (n) | SD (%) | Mean (n) | SD (%) | Mean (n) | SD (%) | |
| Vitamin C, mg/1000 Kcal | 63.04 | 31.90 | 53.24 | 25.05 | 73.39 | 35.07 | <**0.001** |

* *p*-values were calculated from the Student's *t*-tests for continuous variables and from the Chi-square test for categorical variables (*p*-values less than 0.05 are highlighted in bold). Abbreviations: SD, standard deviation; BMI, body mass index; PSQI, Pittsburgh Sleep Quality Index.

### Effects of the interaction between regular exercise and the PSQI on nutrient intake

The regular exercise group was divided into two groups based on PSQI scores; there were 43 participants in the PSQI ≤10 group and 16 in the PSQI ≥11 group. The non-regular exercise group was similarly divided into two groups based on the PSQI scores; there were 63 participants in the PSQI ≤10 group and 63 in the PSQI ≥11 group (Table 4). A two-way ANCOVA adjusting for age, sex, BMI, current smoker, current drinker, education, hypertension, and diabetes was used to examine the effects of interactions between regular exercise and the PSQI on nutrient intake. Interactions were observed for age ($p = 0.006$), education ($p = 0.002$), protein ($p = 0.002$), carbohydrate ($p = 0.045$), phosphorus ($p = 0.008$), zinc ($p = 0.031$), vitamin D ($p = 0.015$), vitamin $B_{12}$ ($p = 0.007$), and pantothenic acid ($p = 0.008$). A post hoc Bonferroni analysis indicated that there was significantly higher protein intake in the PSQI ≤10 group than in the PSQI ≥11 group with regular exercise ($p = 0.001$); however, there was no difference between the two PSQI groups without regular exercise (S1 Fig).

### Effects of regular exercise and protein intake on sleep quality

Table 5 shows the results of a multiple logistic regression analysis with PSQI (≤ 10 and ≥11) stratified by regular exercise. Protein intake was a significant independent variable in any models that were adjusted for individual factors (age, sex, and BMI; OR: 1.260; 95% CI: 1.037, 1.531; $p = 0.020$), individual and environmental factors (current smoker and current drinker; OR: 1.357; 95% CI: 1.081, 1.704; $p = 0.009$), and individual and disease factors (hypertension and diabetes; OR: 1.675; 95% CI: 1.206, 2.326; $p = 0.002$) in the regular exercise group but not in the non-regular exercise group. This result implies that sleep quality is better with a high protein intake, even after adjusting for different confounding factors only in the regular exercise group.

## Discussion

In the present study, the PSQI was selected as the most frequently used index for sleep evaluation. Epidemiological studies on sleep have been performed using sleep times [17, 19] and questionnaires [15, 16, 18, 20, 21]. However, evaluating sleep by time alone lacks objectivity because sleep times and measurement items differ among studies. For example, one study considered the appropriate sleep time to be 7–8 hours [17], but another considered it to be 7–9 hours [19]; other studies have evaluated sleep-related time based on sleep latency (difficulty falling asleep) or sleep efficiency (maintaining sleep) [31]. Therefore, comprehensively evaluating sleep quality using a questionnaire may provide more objective findings. A previous study that compared the diagnostic screening characteristics of the Insomnia Severity Index, the Athens Insomnia Scale, and the PSQI reported similar sensitivities and specificities [32]. Therefore, the PSQI in the present study was confirmed to be a valid and comparable questionnaire to those used in other studies. Buysse *et al.* [28] defined a PSQI score >5 as poor

**Table 2. Differences in characteristics and daily nutrient intake between the PSQI ≤10 and ≥11 groups.**

| | Total (N = 185) | | | | |
|---|---|---|---|---|---|
| | PSQI ≤ 10 (*n* = 106) | | PSQI ≥ 11 (n = 79) | | *p* Value * |
| | Mean (*n*) | SD (%) | Mean (*n*) | SD (%) | |
| Age, years | 59.2 | 9.9 | 62.4 | 9.3 | **0.026** |
| Sex: male, n (%) | 50 | 47.2 | 45 | 57.0 | 0.234 |
| BMI, kg/m$^2$ | 23.4 | 3.1 | 22.6 | 3.0 | 0.064 |
| Exercise / week, days | 1.9 | 2.5 | 1.0 | 1.8 | **0.004** |
| Exercise time / each session, minutes | 35.3 | 61.9 | 16.3 | 32.5 | **0.008** |
| Current smoker, n (%) | 19 | 17.9 | 12 | 15.2 | 0.693 |
| Current drinker, n (%) | 58 | 54.7 | 49 | 62.0 | 0.367 |
| Education | | | | | 0.427 |
| Junior high school, n (%) | 18 | 17.0 | 20 | 25.3 | |
| High school, n (%) | 48 | 45.3 | 31 | 39.2 | |
| Junior college, n (%) | 23 | 21.7 | 16 | 20.3 | |
| University or higher, n (%) | 17 | 16.0 | 12 | 15.2 | |
| PSQI | 8.9 | 1.2 | 13.2 | 2.2 | **<0.001** |
| **Underlying diseases** | | | | | |
| metabolic syndrome, n (%) | 33 | 31.1 | 13 | 16.5 | **0.026** |
| Hypertension, n (%) | 33 | 31.1 | 28 | 35.6 | 0.636 |
| Diabetes, n (%) | 6 | 5.9 | 8 | 10.1 | 0.275 |
| Angina, n (%) | 5 | 5.0 | 5 | 6.8 | 0.746 |
| Myocardial infarction, n (%) | 2 | 2.0 | 1 | 1.4 | 1.000 |
| Depression, n (%) | 0 | 0.0 | 1 | 1.4 | 0.427 |
| **Nutrients** | | | | | |
| Total energy, Kcal | 1993.88 | 673.94 | 1933.77 | 574.35 | 0.524 |
| Protein, %energy | 15.48 | 3.07 | 14.61 | 3.05 | 0.056 |
| Fat, % energy | 25.80 | 5.54 | 24.77 | 6.04 | 0.229 |
| Carbohydrate, % energy | 53.58 | 7.35 | 53.76 | 7.94 | 0.873 |
| minerals, % energy | 10.46 | 1.95 | 10.13 | 2.01 | 0.270 |
| Sodium, mg/1000 Kcal | 2407.42 | 471.59 | 2384.80 | 509.17 | 0.755 |
| Potassium, mg/1000 Kcal | 1449.58 | 418.96 | 1365.07 | 400.18 | 0.168 |
| Calcium, mg/1000 Kcal | 300.81 | 115.94 | 279.39 | 100.56 | 0.190 |
| Magnesium, mg/1000 Kcal | 143.67 | 32.37 | 138.12 | 31.44 | 0.244 |
| Phosphorus, mg/1000 Kcal | 587.05 | 133.72 | 554.28 | 121.50 | 0.088 |
| Iron, mg/1000 Kcal | 4.34 | 1.06 | 4.08 | 1.02 | 0.097 |
| Zinc, mg/1000 Kcal | 4.55 | 0.61 | 4.37 | 0.66 | 0.058 |
| β-carotene equivalent, μg/1000 Kcal | 2151.41 | 1347.64 | 1900.95 | 1172.67 | 0.179 |
| Retinol equivalent, μg/1000 Kcal | 381.22 | 189.21 | 330.20 | 152.28 | **0.044** |
| Vitamin D, μg/1000 Kcal | 8.56 | 5.74 | 7.65 | 4.43 | 0.226 |
| α-Tocopherol, mg/1000 Kcal | 4.02 | 1.05 | 3.89 | 1.07 | 0.398 |
| Vitamin K, μg/1000 Kcal | 175.54 | 93.46 | 158.81 | 77.11 | 0.185 |
| Vitamin B1, mg/1000 Kcal | 0.42 | 0.09 | 0.40 | 0.10 | 0.183 |
| Vitamin B2, mg/1000 Kcal | 0.69 | 0.18 | 0.63 | 0.16 | **0.024** |
| Niacin, mg/1000 Kcal | 9.77 | 2.37 | 9.06 | 2.61 | 0.054 |
| Vitamin B6, mg/1000 Kcal | 0.70 | 0.18 | 0.69 | 0.17 | 0.541 |
| Vitamin B12, μg/1000 Kcal | 5.72 | 2.83 | 5.28 | 2.63 | 0.290 |
| Folic acid, μg/1000 Kcal | 181.94 | 65.37 | 174.14 | 60.32 | 0.408 |
| Pantothenic acid, mg/1000 Kcal | 3.49 | 0.74 | 3.29 | 0.67 | 0.058 |

*(Continued)*

**Table 2.** (Continued)

| | Total (N = 185) | | | | |
| | PSQI ≤ 10 (*n* = 106) | | PSQI ≥ 11 (n = 79) | | *p* Value [*] |
| | Mean (*n*) | SD (%) | Mean (*n*) | SD (%) | |
| Vitamin C, mg/1000 Kcal | 62.66 | 31.54 | 63.56 | 32.57 | 0.850 |

[*] *p*-values were calculated from the Student's *t*-tests for continuous variables and from the Chi-square test for categorical variables (*p*-values less than 0.05 are highlighted in bold). Abbreviations: PSQI, Pittsburgh Sleep Quality Index; SD, standard deviation; BMI, body mass index.

sleep quality. Conversely, Das *et al*. [33] demonstrated that the mean PSQI was 8.59 ± 5.35 in a community-based study among a geriatric population, and they described that the difference in the PSQI may be due to the different cultures and lifestyles of people in different countries. The mean PSQI of all the participants in the present study was 10.7 ± 2.7, which seemed to reflect the current average Japanese lifestyle.

Comparisons between the PSQI ≤10 and ≥11 groups in the present study revealed that regular exercise was beneficial for sleep quality, which is consistent with previous findings [8–12, 34]. However, other studies did not observe a relationship between exercise and sleep [35, 36]. Briefly, the lowest or highest levels of exercise were not associated with sleep disorders [35], and the effects of short-term resistance exercise on sleep were inconsistent [36]. By contrast, previous studies reported a positive relationship between exercise intensity and sleep with 30 minutes or more of exercise each time [8], moderate to intense physical activity of 150 minutes or more per week [10], or 500 to 1500 metabolic equivalents of task minutes/week of physical activity [34]. Exercise intensity in the regular exercise group in the present study was considered intermediate because the mean days of exercise per week was 4.3 (*SD* = 1.7) and the mean exercise time per session was 80.3 minutes (*SD* = 64.2). Accordingly, the relationship observed between exercise intensity and sleep in the present study appears to support previous findings showing that intermediate exercise intensity has a positive effect on sleep quality.

The multiple logistic regression analysis in the present study revealed a positive correlation between good sleep quality and protein intake only in the regular exercise group. These results seem to indicate that there is a mechanism by which regular exercise promotes protein absorption. In addition to exercise, the ingestion of protein just before sleep has been reported to improve nighttime protein synthesis by enhancing its digestion and absorption [37]. Tryptophan is a constituent amino acid of protein that competes with the other larger neutral amino acids to gain access to the transport system to cross the blood-brain barrier. Dietary carbohydrates pull larger amino acids into the muscle tissue, allowing tryptophan to access the transport system, cross the blood-brain barrier, and contribute to the synthesis of serotonin and melatonin [38]. Tryptophan has been shown to affect the serotonin-melatonin pathway because an intraperitoneal injection in rats increased serotonin levels [39]; likewise, its administration to patients with moderate insomnia significantly reduced sleep latency [40]. The reason for good sleep quality among participants who reported high protein intake in their daily diet and regular exercise was thought to be a result of the pathogenesis in which the tryptophan-serotonin-melatonin pathway was activated due to the enhanced protein absorption.

In the present study, the following micronutrients were associated with sleep quality after adjusting for confounding factors (age, sex, BMI, current smoker, current drinker, education, hypertension, and diabetes): phosphorus, zinc, vitamin D, vitamin $B_{12}$, and pantothenic acid. Many of these micronutrients showed similar results as previous studies [16, 17, 31]. Frank *et al*. reported that lower intakes of phosphorus and zinc were associated with a shorter sleep duration [31]. Komada *et al*. [16] reported that vitamin D and vitamin $B_{12}$ in adult males were

**Table 3. Differences in characteristics and daily nutrient intake between regular and non-regular exercise groups.**

| | Total (N = 185) | | | | p Value [*] |
|---|---|---|---|---|---|
| | Regular exercise (*n* = 59) | | Non-regular exercise (*n* = 126) | | |
| | Mean (*n*) | SD (%) | Mean (*n*) | SD (%) | |
| Age, years | 64.2 | 7.8 | 58.8 | 10.1 | **<0.001** |
| Sex: male, n (%) | 26 | 44.1 | 69 | 54.8 | 0.208 |
| BMI, kg/m² | 22.8 | 3.0 | 23.2 | 3.1 | 0.378 |
| Exercise / week, days | 4.3 | 1.7 | 0.2 | 0.8 | **<0.001** |
| Exercise time / each session, minutes | 80.3 | 64.2 | 2.3 | 11.7 | **<0.001** |
| Current smoker, n (%) | 5 | 8.5 | 26 | 20.6 | 0.056 |
| Current drinker, n (%) | 30 | 50.8 | 77 | 61.1 | 0.204 |
| Education | | | | | 0.536 |
| Junior high school, n (%) | 13 | 22.0 | 25 | 19.8 | |
| High school, n (%) | 27 | 45.8 | 52 | 41.3 | |
| Junior college, n (%) | 10 | 16.9 | 29 | 23.0 | |
| University or higher, n (%) | 9 | 15.3 | 20 | 15.9 | |
| PSQI | 9.9 | 2.2 | 11.1 | 2.9 | **0.003** |
| **Underlying diseases** | | | | | |
| metabolic syndrome, n (%) | 15 | 25.4 | 31 | 24.6 | 1.000 |
| Hypertension, n (%) | 21 | 35.6 | 40 | 31.7 | 0.618 |
| Diabetes, n (%) | 7 | 11.9 | 7 | 5.6 | 0.144 |
| Angina, n (%) | 2 | 3.4 | 8 | 6.3 | 0.506 |
| Myocardial infarction, n (%) | 1 | 1.7 | 2 | 1.6 | 1.000 |
| Depression, n (%) | 0 | 0.0 | 1 | 0.8 | 1.000 |
| **Nutrients** | | | | | |
| Total energy, Kcal | 1961.70 | 554.63 | 1971.26 | 667.77 | 0.924 |
| Protein, %energy | 16.33 | 3.42 | 14.54 | 2.74 | **<0.001** |
| Fat, % energy | 25.79 | 6.04 | 25.16 | 5.64 | 0.488 |
| Carbohydrate, % energy | 52.63 | 6.80 | 54.14 | 7.90 | 0.206 |
| minerals, % energy | 11.10 | 1.99 | 9.95 | 1.87 | **<0.001** |
| Sodium, mg/1000 Kcal | 2460.96 | 473.81 | 2368.17 | 491.78 | 0.228 |
| Potassium, mg/1000 Kcal | 1624.20 | 432.14 | 1314.83 | 364.07 | **<0.001** |
| Calcium, mg/1000 Kcal | 349.42 | 124.82 | 264.62 | 90.70 | **<0.001** |
| Magnesium, mg/1000 Kcal | 156.34 | 32.86 | 134.25 | 29.15 | **<0.001** |
| Phosphorus, mg/1000 Kcal | 635.65 | 146.91 | 543.75 | 109.04 | **<0.001** |
| Iron, mg/1000 Kcal | 4.71 | 1.10 | 4.00 | 0.95 | **<0.001** |
| Zinc, mg/1000 Kcal | 4.72 | 0.72 | 4.36 | 0.57 | **<0.001** |
| β-carotene equivalent, μg/1000 Kcal | 2629.14 | 1537.63 | 1770.68 | 1035.57 | **<0.001** |
| Retinol equivalent, μg/1000 Kcal | 436.79 | 206.07 | 323.21 | 147.08 | **<0.001** |
| Vitamin D, μg/1000 Kcal | 9.95 | 6.46 | 7.34 | 4.32 | **0.001** |
| α-Tocopherol, mg/1000 Kcal | 4.34 | 1.16 | 3.79 | 0.97 | **0.001** |
| Vitamin K, μg/1000 Kcal | 199.24 | 92.18 | 153.95 | 80.91 | **0.001** |
| Vitamin B1, mg/1000 Kcal | 0.46 | 0.10 | 0.39 | 0.08 | **<0.001** |
| Vitamin B2, mg/1000 Kcal | 0.75 | 0.16 | 0.63 | 0.16 | **<0.001** |
| Niacin, mg/1000 Kcal | 10.21 | 2.54 | 9.12 | 2.40 | **0.005** |
| Vitamin B6, mg/1000 Kcal | 0.77 | 0.17 | 0.66 | 0.16 | **<0.001** |
| Vitamin B12, μg/1000 Kcal | 6.18 | 2.92 | 5.23 | 2.62 | **0.027** |
| Folic acid, μg/1000 Kcal | 207.76 | 66.76 | 164.96 | 56.79 | **<0.001** |
| Pantothenic acid, mg/1000 Kcal | 3.74 | 0.73 | 3.25 | 0.66 | **<0.001** |

(*Continued*)

**Table 3.** (Continued)

| | Total (N = 185) | | | | p Value * |
|---|---|---|---|---|---|
| | Regular exercise (*n* = 59) | | Non-regular exercise (*n* = 126) | | |
| Vitamin C, mg/1000 Kcal | 76.62 | 34.64 | 56.68 | 28.52 | <**0.001** |

* *p*-values were calculated from the Student's *t*-tests for continuous variables and from the Chi-square test for categorical variables (*p*-values less than 0.05 are highlighted in bold). Abbreviations: PSQI, Pittsburgh Sleep Quality Index; SD, standard deviation; BMI, body mass index.

associated with sleep duration. Grandner *et al.* [17] reported that vitamin D was associated with sleep maintenance difficulties. Since the results for many of the micronutrients examined in the present study agreed with those of previous studies, the relationships observed between micronutrient intakes and sleep in the present study were considered reliable.

**Table 4. Interactions between exercise groups and PSQI groups.**

| | Total (*N* = 185) | | | | | | | | | | | | p Value * | | |
|---|---|---|---|---|---|---|---|---|---|---|---|---|---|---|---|
| | Regular exercise (*n* = 59) | | | | | | Non-regular exercise (*n* = 126) | | | | | | RE | PSQI | RE * PSQI |
| | PSQI ≤ 10 (n = 43) | | | PSQI ≥ 11 (n = 16) | | | PSQI ≤ 10 (n = 63) | | | PSQI ≥ 11 (n = 63) | | | | | |
| | Mean | 95% CI | | Mean | 95% CI | | Mean | 95% CI | | Mean | 95% CI | | | | |
| | | Lower | Upper | | Lower | Upper | | Lower | Upper | | Lower | Upper | | | |
| Age, years | 64.2 | 61.9 | 66.6 | 64.2 | 59.8 | 68.6 | 55.7 | 53.3 | 58.2 | 61.9 | 59.5 | 64.3 | **0.003** | 0.180 | **0.006** |
| Sex † | 1.4 | 1.2 | 1.6 | 1.6 | 1.3 | 1.8 | 1.5 | 1.4 | 1.7 | 1.6 | 1.5 | 1.7 | 0.532 | 0.276 | 0.469 |
| BMI, kg/m² | 23.0 | 22.1 | 23.8 | 22.2 | 20.1 | 24.2 | 23.7 | 22.9 | 24.6 | 22.7 | 22.0 | 23.4 | 0.217 | **0.037** | 0.721 |
| Current smoker ‡ | 1.9 | 1.8 | 2.0 | 1.9 | 1.8 | 2.1 | 1.8 | 1.7 | 1.9 | 1.8 | 1.7 | 1.9 | 0.111 | 0.2 | 0.702 |
| Current drinker ‡ | 1.5 | 1.4 | 1.7 | 1.4 | 1.1 | 1.6 | 1.4 | 1.3 | 1.5 | 1.4 | 1.3 | 1.5 | 0.525 | 0.279 | 0.751 |
| Education § | 2.4 | 2.1 | 2.7 | 1.9 | 1.4 | 2.4 | 2.4 | 2.1 | 2.6 | 2.4 | 2.1 | 2.6 | 0.712 | 0.414 | **0.002** |
| **Underlying diseases ¶** | | | | | | | | | | | | | | | |
| Metabolic syndrome, % | 1.7 | 1.6 | 1.86 | 1.81 | 1.6 | 2.0 | 1.7 | 1.6 | 1.8 | 1.8 | 1.8 | 1.9 | 0.565 | 0.133 | 0.922 |
| Hypertension, % | 1.6 | 1.5 | 1.78 | 1.69 | 1.4 | 1.9 | 1.7 | 1.6 | 1.8 | 1.6 | 1.5 | 1.8 | 0.566 | 0.601 | 0.332 |
| Diabetes, % | 1.9 | 1.9 | 2.01 | 1.75 | 1.5 | 2.0 | 2.0 | 1.9 | 2.0 | 1.9 | 1.9 | 2.0 | **0.013** | **0.023** | 0.064 |
| Angina, % | 2.0 | 1.9 | 2.02 | 1.94 | 1.8 | 2.1 | 1.9 | 1.9 | 2.0 | 1.9 | 1.9 | 2.0 | 0.429 | 0.89 | 0.334 |
| Myocardial infarction, % | 2.0 | 1.9 | 2.02 | 2.00 | 2.0 | 2.0 | 2.0 | 2.0 | 2.0 | 2.0 | 2.0 | 2.0 | 0.488 | 0.375 | 0.441 |
| Depression, % | 2.0 | 2.0 | 2.0 | 2.00 | 2.0 | 2.0 | 2.0 | 2.0 | 2.0 | 2.0 | 2.0 | 2.0 | 0.263 | 0.964 | 0.263 |
| **Nutrients** | | | | | | | | | | | | | | | |
| Total energy, Kcal | 2012.22 | 1850.69 | 2173.76 | 1825.92 | 1492.71 | 2159.12 | 1981.37 | 1789.21 | 2173.52 | 1961.16 | 1819.44 | 2102.88 | 0.369 | **0.043** | 0.348 |
| Protein, %energy | 17.13 | 16.11 | 18.14 | 14.19 | 12.65 | 15.73 | 14.36 | 13.77 | 14.95 | 14.71 | 13.93 | 15.49 | 0.321 | **0.005** | **0.002** |
| Fat, % energy | 26.67 | 24.77 | 28.57 | 23.45 | 20.71 | 26.20 | 25.22 | 23.95 | 26.48 | 25.11 | 23.54 | 26.68 | 0.817 | 0.358 | 0.218 |
| Carbohydrate, % energy | 51.59 | 49.34 | 53.84 | 55.42 | 53.16 | 57.67 | 54.94 | 53.15 | 56.73 | 53.34 | 51.17 | 55.51 | 0.610 | 0.317 | **0.045** |
| minerals, % energy | 11.42 | 10.84 | 11.99 | 10.24 | 9.10 | 11.38 | 9.80 | 9.36 | 10.24 | 10.10 | 9.60 | 10.61 | 0.112 | 0.197 | 0.108 |
| Sodium, mg/1000 Kcal | 2523.05 | 2384.08 | 2662.02 | 2294.09 | 2024.30 | 2563.89 | 2328.50 | 2209.59 | 2447.41 | 2407.83 | 2279.05 | 2536.62 | 0.926 | 0.334 | 0.102 |
| Potassium, mg/1000 Kcal | 1671.49 | 1550.93 | 1792.05 | 1497.12 | 1220.88 | 1773.37 | 1298.12 | 1205.30 | 1390.94 | 1331.53 | 1240.45 | 1422.62 | **0.002** | 0.478 | 0.591 |
| Calcium, mg/1000 Kcal | 368.18 | 329.05 | 407.30 | 298.99 | 242.42 | 355.56 | 254.83 | 234.49 | 275.17 | 274.41 | 249.38 | 299.43 | **0.019** | 0.068 | 0.096 |
| Magnesium, mg/1000 Kcal | 161.14 | 151.71 | 170.57 | 143.46 | 124.24 | 162.68 | 131.74 | 124.70 | 138.78 | 136.76 | 129.12 | 144.40 | **0.028** | 0.206 | 0.150 |
| Phosphorus, mg/1000 Kcal | 665.26 | 620.65 | 709.87 | 556.08 | 490.07 | 622.09 | 533.67 | 509.90 | 557.44 | 553.83 | 523.13 | 584.53 | 0.058 | **0.018** | **0.008** |
| Iron, mg/1000 Kcal | 4.90 | 4.58 | 5.23 | 4.18 | 3.61 | 4.75 | 3.95 | 3.73 | 4.17 | 4.05 | 3.80 | 4.31 | 0.088 | **0.032** | 0.105 |
| Zinc, mg/1000 Kcal | 4.85 | 4.65 | 5.06 | 4.35 | 3.94 | 4.76 | 4.34 | 4.22 | 4.46 | 4.37 | 4.21 | 4.53 | 0.235 | **0.021** | **0.031** |

*(Continued)*

**Table 4.** (Continued)

| | Total (N = 185) | | | | | | | | | | | | | | | |
| --- | --- | --- | --- | --- | --- | --- | --- | --- | --- | --- | --- | --- | --- | --- | --- | --- |
| | Regular exercise (n = 59) | | | | | | Non-regular exercise (n = 126) | | | | | | p Value * | | |
| | PSQI ≤ 10 (n = 43) | | | PSQI ≥ 11 (n = 16) | | | PSQI ≤ 10 (n = 63) | | | PSQI ≥ 11 (n = 63) | | | RE | PSQI | RE * PSQI |
| | Mean | 95% CI | | Mean | 95% CI | | Mean | 95% CI | | Mean | 95% CI | | | | |
| | | Lower | Upper | | Lower | Upper | | Lower | Upper | | Lower | Upper | | | |
| β-carotene equivalent, μg/1000 Kcal | 2739.29 | 2267.63 | 3210.94 | 2333.12 | 1501.24 | 3165.01 | 1750.17 | 1488.28 | 2012.05 | 1791.19 | 1529.47 | 2052.91 | **0.014** | 0.353 | 0.949 |
| Retinol equivalent, μg/1000 Kcal | 466.74 | 398.27 | 535.20 | 356.31 | 288.61 | 424.01 | 322.85 | 288.53 | 357.17 | 323.57 | 283.71 | 363.43 | **0.028** | **0.047** | 0.208 |
| Vitamin D, μg/1000 Kcal | 10.97 | 8.85 | 13.10 | 7.19 | 5.00 | 9.38 | 6.92 | 5.89 | 7.95 | 7.77 | 6.63 | 8.91 | 0.414 | **0.029** | **0.015** |
| α-Tocopherol, mg/1000 Kcal | 4.51 | 4.16 | 4.85 | 3.87 | 3.25 | 4.49 | 3.69 | 3.47 | 3.91 | 3.89 | 3.63 | 4.16 | 0.181 | 0.402 | 0.13 |
| Vitamin K, μg/1000 Kcal | 214.78 | 185.43 | 244.14 | 157.46 | 120.42 | 194.50 | 148.75 | 127.93 | 169.57 | 159.16 | 139.15 | 179.16 | 0.33 | 0.058 | 0.072 |
| Vitamin B1, mg/1000 Kcal | 0.47 | 0.44 | 0.50 | 0.42 | 0.36 | 0.49 | 0.39 | 0.37 | 0.41 | 0.40 | 0.38 | 0.42 | **0.012** | 0.387 | 0.302 |
| Vitamin B2, mg/1000 Kcal | 0.78 | 0.73 | 0.83 | 0.65 | 0.58 | 0.72 | 0.62 | 0.58 | 0.67 | 0.63 | 0.59 | 0.67 | **0.031** | **0.021** | 0.094 |
| Niacin, mg/1000 Kcal | 10.60 | 9.84 | 11.35 | 9.17 | 7.82 | 10.52 | 9.21 | 8.67 | 9.75 | 9.03 | 8.37 | 9.70 | 0.146 | 0.14 | 0.179 |
| Vitamin B6, mg/1000 Kcal | 0.80 | 0.75 | 0.85 | 0.71 | 0.62 | 0.80 | 0.64 | 0.60 | 0.68 | 0.68 | 0.64 | 0.72 | **0.025** | 0.501 | 0.109 |
| Vitamin B12, μg/1000 Kcal | 6.79 | 5.85 | 7.72 | 4.57 | 3.61 | 5.52 | 4.99 | 4.37 | 5.60 | 5.46 | 4.76 | 6.17 | 0.965 | **0.034** | **0.007** |
| Folic acid, μg/1000 Kcal | 214.72 | 194.79 | 234.65 | 189.06 | 151.45 | 226.67 | 159.57 | 145.45 | 173.69 | 170.35 | 155.88 | 184.82 | **0.02** | 0.425 | 0.449 |
| Pantothenic acid, mg/1000 Kcal | 3.91 | 3.70 | 4.12 | 3.28 | 2.94 | 3.62 | 3.20 | 3.04 | 3.36 | 3.29 | 3.12 | 3.46 | **0.046** | **0.026** | **0.008** |
| Vitamin C, mg/1000 Kcal | 76.90 | 67.10 | 86.70 | 75.86 | 53.24 | 98.47 | 52.93 | 45.99 | 59.88 | 60.43 | 53.09 | 67.78 | **0.008** | 0.382 | 0.731 |

* Analysis of covariance (p-values less than 0.05 are highlighted in bold). Adjusted for age, sex, BMI, current smoker, current drinker, education, hypertension, and diabetes.

† sex (1. female, 2. male)

‡ current smoker or drinker (1. yes, 2. no)

§ education (1. junior high school, 2. high school, 3. junior college, 4. university and higher)

₲ underlying diseases (1. yes, 2. no). Abbreviations: PSQI, Pittsburgh Sleep Quality Index; RE, regular exercise; CI, confidence interval; BMI, body mass index.

**Table 5. Relationship between protein intake and good sleep quality stratified by regular exercise.**

| | | β | p—Value | OR | 95% CI | |
| --- | --- | --- | --- | --- | --- | --- |
| | | | | | Lower | Upper |
| Regular exercise | Model 1 | 0.231 | **0.020** | 1.260 | 1.037 | 1.531 |
| | Model 2 | 0.305 | **0.009** | 1.357 | 1.081 | 1.704 |
| | Model3 | 0.516 | **0.002** | 1.675 | 1.206 | 2.326 |
| Non-regular exercise | Model 1 | -0.010 | 0.880 | 0.990 | 0.870 | 1.127 |
| | Model 2 | -0.006 | 0.925 | 0.994 | 0.868 | 1.137 |
| | Model 3 | -0.035 | 0.624 | 0.966 | 0.840 | 1.110 |

Significant estimates are in bold. Model 1: adjusted for age, sex, and BMI; Model 2: adjusted for age, sex, BMI, current smoker, and current drinker; Model 3: adjusted for age, sex, BMI, hypertension, and diabetes. Abbreviations: β, coefficient; OR, odds ratio; CI, confidence interval; BMI, body mass index.

One limitation of the present study is that the number of inputs for the independent variables was restricted in the multivariate analysis because of the small number of participants. This study might include selection bias because the subjects participated voluntarily in this study. Since the PSQI was evaluated only with a questionnaire, using a more objective method such as a polysomnogram to assess sleep quality is necessary. Since we have not examined the effects of obstructive sleep apnea, future studies should be performed that incorporate a design to evaluate obstructive sleep apnea. Moreover, since this was a cross-sectional study, interventions analyzing regular exercise and protein intake could not be conducted. Further multicenter randomized controlled trials with target values for exercise intensity and protein intake are necessary to clarify the effects of regular exercise and nutrient intake on sleep quality.

## Conclusions

We conducted this cross-sectional study on Japanese participants to investigate the relationship between regular exercise and nutrient intake as factors affecting sleep quality. Protein intake was higher among the participants with a PSQI ≤10 in the regular exercise group (mean, 17.13% of energy consumption) than in those in the non-regular exercise or PSQI ≥11 groups. Furthermore, the results of the multiple logistic regression analysis showed that sleep quality was better in the regular exercise group when protein intake was high; this relationship was not observed in the non-regular exercise group.

## Supporting information

**S1 Fig. Interaction between regular exercise and the PSQI on protein intake.** * Post hoc Bonferroni analysis. Adjusted for age = 60.54, sex = 1.51, BMI = 23.05, current smoker = 1.83, current drinker = 1.42, education = 2.32, hypertension = 1.67, diabetes = 1.92. Error bar: 95% CI. Abbreviations: PSQI, Pittsburgh Sleep Quality Index, EMMEANS, estimated marginal means.
(TIF)

## Acknowledgments

We would like to thank the officials of Shika Town, Ishikawa prefecture and the staff of the Department of Environmental and Preventive Medicine, Kanazawa University Graduate School of Medical Sciences, Kanazawa University Graduate School of Advanced Preventive Medical Sciences, Department of Bioinformatics and Genomics, University of Tsukuba and Keio University.

## Author Contributions

**Conceptualization:** Fumihiko Suzuki, Koichiro Hayashi, Hiroyuki Nakamura.

**Data curation:** Fumihiko Suzuki, Emi Morita, Sakae Miyagi, Hiromasa Tsujiguchi, Akinori Hara, Thao Thi Thu Nguyen, Yukari Shimizu, Koichiro Hayashi, Keita Suzuki, Sumire Matsumoto, Asuka Ishihara, Daisuke Hori, Shotaro Doki, Yuichi Oi, Hiroyuki Nakamura.

**Formal analysis:** Fumihiko Suzuki, Sakae Miyagi, Hiromasa Tsujiguchi, Akinori Hara, Thao Thi Thu Nguyen, Yukari Shimizu, Koichiro Hayashi, Keita Suzuki, Hiroyuki Nakamura.

**Funding acquisition:** Makoto Satoh, Hiroyuki Nakamura.

**Investigation:** Fumihiko Suzuki, Sakae Miyagi, Hiromasa Tsujiguchi, Akinori Hara, Thao Thi Thu Nguyen, Yukari Shimizu, Koichiro Hayashi, Keita Suzuki, Hiroyuki Nakamura.

**Project administration:** Fumihiko Suzuki, Hiroyuki Nakamura.

**Resources:** Hiroyuki Nakamura.

**Supervision:** Hiroyuki Nakamura.

**Validation:** Fumihiko Suzuki, Emi Morita, Sakae Miyagi, Hiromasa Tsujiguchi, Akinori Hara, Thao Thi Thu Nguyen, Yukari Shimizu, Koichiro Hayashi, Keita Suzuki, Takayuki Kannon, Atsushi Tajima, Sumire Matsumoto, Asuka Ishihara, Daisuke Hori, Shotaro Doki, Yuichi Oi, Shinichiro Sasahara, Makoto Satoh, Ichiyo Matsuzaki, Masashi Yanagisawa, Toshiharu Ikaga, Hiroyuki Nakamura.

**Writing – original draft:** Fumihiko Suzuki.

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
