## [Decision Letter · Decision Letter 0]

24 Nov 2020

PONE-D-20-33985

Protein intake in inhabitants with regular exercise is associated with sleep quality: Result of the Shika study

PLOS ONE

Dear Dr. Suzuki,

Thank you for submitting your manuscript to PLOS ONE. After careful consideration, we feel that it has merit but does not fully meet PLOS ONE’s publication criteria as it currently stands. Therefore, we invite you to submit a revised version of the manuscript that addresses the points raised during the review process.

It is necessary that the authors address the requirements expressed by the reviewers.

We look forward to receiving your revised manuscript.

Kind regards,

Jose M. Moran

Academic Editor

PLOS ONE

Journal Requirements:

2. In your Methods section, please provide additional information about the participant recruitment method and the demographic details of your participants. Please ensure you have provided sufficient details to replicate the analyses such as: a) the recruitment date range (month and year), b) a description of any inclusion/exclusion criteria that were applied to participant recruitment, c) a table of relevant demographic details, d) a statement as to whether your sample can be considered representative of a larger population, e) a description of how participants were recruited, and f) descriptions of where participants were recruited and where the research took place. Finally, please report whether sample size calculations were performed.

3. Please include additional information regarding the survey or questionnaire used in the study and ensure that you have provided sufficient details that others could replicate the analyses. For instance, if you developed a questionnaire as part of this study and it is not under a copyright more restrictive than CC-BY, please include a copy, in both the original language and English, as Supporting Information. Moreover, please include more details on how the questionnaire was pre-tested, and whether it was validated.

Reviewers' comments:

Reviewer's Responses to Questions

**Comments to the Author**

1. Is the manuscript technically sound, and do the data support the conclusions?

Reviewer #1: Yes

Reviewer #2: Yes

2. Has the statistical analysis been performed appropriately and rigorously? 

Reviewer #1: Yes

Reviewer #2: Yes

3. Have the authors made all data underlying the findings in their manuscript fully available?

Reviewer #1: Yes

Reviewer #2: Yes

4. Is the manuscript presented in an intelligible fashion and written in standard English?

Reviewer #1: Yes

Reviewer #2: Yes

5. Review Comments to the Author

Reviewer #1: I have read with great interest the study of Fumihiko Suzuki and collaborators and I want to comment the following:

1. I am struck by the division of two study groups based on a cut-off point of 10 in the PSQI without a clinical justification.

The original PSQI study concluded that a value greater than 5 indicates a negative impact on sleep. This data is mentioned by the authors in the methodology. However, it makes no mention of the Japanese validation study that found similar data. That is, values >5.5 are indicative of poor sleep quality. By the way, in the study the control group had an average 38 years old and a cut-off point of 7.5, a specificity of 97% was found for those patients with primary insomnia. Then, an average of 10 points seems to be high and difficult to be considered normal. (Doi Y, Minowa M, Uchiyama M, Okawa M, Kim K, Shibui K, et al. Psychometric assessment of subjective sleep quality using the Japanese version of the Pittsburgh Sleep Quality Index (PSQI-J) in psychiatric disordered and control subjects. Psychiatry Res. 2000;97: 165–172.)

The authors could use the cutoff points that best fit their purposes, but as a post-frame analysis. In strict adherence to the above, it would be desirable for the researchers to add an analysis considering the cut-off point of> 5.5 and continue with the groups they consider.

2. The discussion should mention the possible causes for which the average population has poor sleep quality (mean PSQI of 10) and be careful to consider the group with <10 as having a good quality of sleep.

Reviewer #2: the manuscript is well written with novel findings

comments:

methodology:

the author needs to define what is PSQI and BDHQ and did you use Japanese version and if so, any validation study done

Results:

good analysis was done

Discussion:

PSQI has its own drawback for sleep quality assessment. furthermore, people with metabolic syndrome are at greater risk of developing obstructive sleep apnea and that may hinder their sleep quality

please add these comments in the discussion

6. PLOS authors have the option to publish the peer review history of their article (what does this mean?). If published, this will include your full peer review and any attached files.

Reviewer #1: No

Reviewer #2: No

---

## [Author Response · Author response to Decision Letter 0]

14 Jan 2021

I want my identity to be published for this peer review.

---

## [Decision Letter · Decision Letter 1]

11 Feb 2021

PONE-D-20-33985R1

Protein intake in inhabitants with regular exercise is associated with sleep quality: Result of the Shika study

PLOS ONE

Dear Dr. Suzuki,

Thank you for submitting your manuscript to PLOS ONE. After careful consideration, we feel that it has merit but does not fully meet PLOS ONE’s publication criteria as it currently stands. Therefore, we invite you to submit a revised version of the manuscript that addresses the points raised during the review process.

Although authors have appropriately addressed the requirements of the reviewers, before recommending publication of the article, and given that the authors have included in their analysis the t-Student, ANCOVA, multiple linear regression and Pearson correlations, the authors should state in the methodology section that all the variables involved fulfilled the assumptions required by parametric methods and that these were tested by the appropriate tests (indicate which ones the authors have used).

We look forward to receiving your revised manuscript.

Kind regards,

Jose M. Moran

Academic Editor

PLOS ONE

Reviewers' comments:

Reviewer's Responses to Questions

**Comments to the Author**

1. If the authors have adequately addressed your comments raised in a previous round of review and you feel that this manuscript is now acceptable for publication, you may indicate that here to bypass the “Comments to the Author” section, enter your conflict of interest statement in the “Confidential to Editor” section, and submit your "Accept" recommendation.

Reviewer #1: All comments have been addressed

Reviewer #2: All comments have been addressed

2. Is the manuscript technically sound, and do the data support the conclusions?

Reviewer #1: Yes

Reviewer #2: Yes

3. Has the statistical analysis been performed appropriately and rigorously? 

Reviewer #1: Yes

Reviewer #2: I Don't Know

4. Have the authors made all data underlying the findings in their manuscript fully available?

Reviewer #1: Yes

Reviewer #2: Yes

5. Is the manuscript presented in an intelligible fashion and written in standard English?

Reviewer #1: Yes

Reviewer #2: Yes

6. Review Comments to the Author

Reviewer #1: The authors' response to the observation of the cut-off points is valid because they are supported by a recent report on the geriatric population and therefore responds favorably to the question about the high value of the Pittsburgh sleep quality scale in the population of this study.

Reviewer #2: minor linguistic errors,

i would suggest writing a review linking, sleep, exercise fitness and diet. the focus on sleep not only from OSA prospective but rather descriping the link between sleep duration, timing and physical fitness in addition to healthy diet

7. PLOS authors have the option to publish the peer review history of their article (what does this mean?). If published, this will include your full peer review and any attached files.

Reviewer #1: No

Reviewer #2: **Yes: **Mohammed A. Al-Abri

---

## [Author Response · Author response to Decision Letter 1]

16 Feb 2021

Feb 16th, 2021.

Jose M. Moran, Ph. D

Academic Editor

PLOS ONE

Dear Dr. Moran,

Thank you for inviting us to submit a revised draft of our manuscript entitled, “Protein intake in inhabitants with regular exercise is associated with sleep quality: Results of the Shika study” to PLOS ONE. We also appreciate the time and effort you and each of the reviewers have dedicated to providing insightful feedback on ways to strengthen our paper. Thus, it is with great pleasure that we resubmit our article for further consideration. We have incorporated changes that reflect the detailed suggestions you have graciously provided. We also hope that our edits and the responses we provide below satisfactorily address all the issues and concerns you and the reviewers have noted.

To facilitate your review of our revisions, the following are our point-by-point responses to the questions and comments delivered in your letter dated Feb 11th, 2020.

Major points.

Q1. Although authors have appropriately addressed the requirements of the reviewers, before recommending publication of the article, and given that the authors have included in their analysis the t-Student, ANCOVA, multiple linear regression and Pearson correlations, the authors should state in the methodology section that all the variables involved fulfilled the assumptions required by parametric methods and that these were tested by the appropriate tests (indicate which ones the authors have used).

A1. We have added the following phrase to the revised manuscript: “The distribution of variables was checked by the Kolmogorov–Smirnov, and Shapiro–Wilk normality tests, or the normal distribution curve in the histogram was confirmed before using other statistical tests.” (P9 L180–182).

Supplementary explanation: Although some variables had a p-value of 0.05 or less in the Kolmogorov–Smirnov, and Shapiro–Wilk normality test, it was confirmed that a normal distribution curve was drawn in the histogram. Therefore, we believe that the variables used in this study can be expected to have a multivariate normal distribution.

Reviewer #1:

There were no additional comments.

Reviewer #2

Q1. Has the statistical analysis been performed appropriately and rigorously? (Reviewer #2: I Don't Know)

A1. We have added the following phrase to the revised manuscript: “The distribution of variables was checked by the Kolmogorov–Smirnov, and Shapiro–Wilk normality tests, or the normal distribution curve in the histogram was confirmed before using other statistical tests.” (P9 L180–182).

Supplementary explanation: Although some variables had a p-value of 0.05 or less in the Kolmogorov–Smirnov, and Shapiro–Wilk normality test, it was confirmed that a normal distribution curve was drawn in the histogram. Therefore, we believe that the variables used in this study can be expected to have a multivariate normal distribution.

Q2. minor linguistic errors,

i would suggest writing a review linking, sleep, exercise fitness and diet. the focus on sleep not only from OSA prospective but rather descriping the link between sleep duration, timing and physical fitness in addition to healthy diet.

A2. Since we were conducting a cross-sectional analysis, we cannot examine the sleep duration and timing in detail, but the mechanism by which regular exercise and nutrition on sleep quality were described as follows: "The reason for good sleep quality among participants who reported high protein intake in their daily diet and regular exercise was thought to be a result of the pathogenesis in which the tryptophan-serotonin-melatonin pathway was activated due to the enhanced protein absorption" (P20 L348-351).

Again, thank you for giving us the opportunity to strengthen our manuscript with your valuable comments and queries. We have worked hard to incorporate your feedback and hope that these revisions persuade you to accept our submission.

Sincerely,

Fumihiko Suzuki

Department of Environmental and Preventive Medicine, Graduate School of Medical Science, Kanazawa University, 13-1 Takaramachi, Kanazawa, Ishikawa 920-8640, Japan.

Tel: +81-76-265-2218

Email address: f-suzuki@stu.kanazawa-u.ac.jp

---

## [Editor Report · Decision Letter 2]

17 Feb 2021

Protein intake in inhabitants with regular exercise is associated with sleep quality: Result of the Shika study

PONE-D-20-33985R2

Dear Dr. Suzuki,

We’re pleased to inform you that your manuscript has been judged scientifically suitable for publication and will be formally accepted for publication once it meets all outstanding technical requirements.

Kind regards,

Jose M. Moran

Academic Editor

PLOS ONE
---

## [Editor Report · Acceptance letter]

19 Feb 2021

PONE-D-20-33985R2 

Protein intake in inhabitants with regular exercise is associated with sleep quality: Results of the Shika study 

Dear Dr. Suzuki:

I'm pleased to inform you that your manuscript has been deemed suitable for publication in PLOS ONE. Congratulations! Your manuscript is now with our production department. 

Kind regards, 

on behalf of

Dr. Jose M. Moran 

Academic Editor

PLOS ONE